# Urinary Sodium Excretion and Blood Pressure Relationship across Methods of Evaluating the Completeness of 24-h Urine Collections

**DOI:** 10.3390/nu12092772

**Published:** 2020-09-11

**Authors:** Abu Mohd Naser, Feng J. He, Mahbubur Rahman, K. M. Venkat Narayan, Norm R. C. Campbell

**Affiliations:** 1Emory Global Diabetes Research Center, Hubert Department of Global Health, Rollins School of Public Health, Emory University, Atlanta, GA 30322, USA; KNARAYA@emory.edu; 2Department of Epidemiology, Rollins School of Public Health, Emory University, Atlanta, GA 30322, USA; 3Centre for Environmental and Preventive Medicine, Wolfson Institute of Preventive Medicine, Barts and The London School of Medicine and Dentistry, Queen Mary University of London, London E1 2AD, UK; f.he@qmul.ac.uk; 4International Centre for Diarrhoeal Disease Research, Dhaka 1212, Bangladesh; mahbubr@icddrb.org; 5Department of Medicine, O’Brien Institute of Public Health, Libin Cardiovascular Institute of Alberta at the University of Calgary, Calgary, AB T2N 4Z6, Canada; ncampbel@ucalgary.ca

**Keywords:** blood pressure, sodium, salt intake, urinary sodium, complete 24-h urine

## Abstract

We compared the sodium intake and systolic blood pressure (SBP) relationship from complete 24-h urine samples determined by several methods: self-reported no-missed urine, creatinine index ≥0.7, measured 24-h urine creatinine (mCER) within 25% and 15% of Kawasaki predicted urine creatinine, and sex-specific mCER ranges (mCER 15–25 mg/kg/24-h for men; 10–20 mg/kg/24-h for women). We pooled 10,031 BP and 24-h urine sodium data from 2143 participants. We implemented multilevel linear models to illustrate the shape of the sodium–BP relationship using the restricted cubic spline (RCS) plots, and to assess the difference in mean SBP for a 100 mmol increase in 24-h urine sodium. The RCS plot illustrated an initial steep positive sodium–SBP relationship for all methods, followed by a less steep positive relationship for self-reported no-missed urine, creatinine index ≥0.7, and sex-specific mCER ranges; and a plateaued relationship for the two Kawasaki methods. Each 100 mmol/24-h increase in urinary sodium was associated with 0.64 (95% CI: 0.34, 0.94) mmHg higher SBP for self-reported no-missed urine, 0.68 (95% CI: 0.27, 1.08) mmHg higher SBP for creatinine index ≥0.7, 0.87 (95% CI: 0.07, 1.67) mmHg higher SBP for mCER within 25% Kawasaki predicted urine creatinine, 0.98 (95% CI: −0.07, 2.02) mmHg change in SBP for mCER within 15% Kawasaki predicted urine creatinine, and 1.96 (95% CI: 0.93, 2.99) mmHg higher SBP for sex-specific mCER ranges. Studies examining 24-h urine sodium in relation to health outcomes will have different results based on how urine collections are deemed as complete.

## 1. Introduction

High systolic blood pressure (SBP) is the largest single risk for death globally [1]. High intake of sodium is an important dietary risk factor for high blood pressure (BP) [2]. High-quality epidemiological studies and systematic reviews show strong evidence that high sodium intake causes high BP and there is moderately strong evidence from clinical trials for causing cardiovascular diseases [3,4,5,6]. In general, the current diets of populations across the globe contain much higher than recommended sodium [7,8]. Long-term reduction in sodium intake is one of the most cost-effective strategies to reduce blood pressure and cardiovascular disease [9].

Accurate measurement of daily sodium intake is critical to evaluate the effectiveness of population-based sodium reduction interventions [3]. Sodium measurement in 24-h urine samples is the recommended method to assess daily sodium intake since 93% of ingested sodium is excreted within 24-h [10]. However, 24-h urine collections are often affected by under- or over-collection, which can result in erroneous estimation of sodium intake [11]. Inclusion of under- or over-collected 24-h urine samples in epidemiological studies may also result in a spurious relationship between sodium intake and health outcomes. Some epidemiological studies use urinary para-aminobenzoic acid (PABA), urinary creatinine, and self-reports for assessment of the completeness of 24-h urine collection [12]. The urinary PABA recovery method is a recommended approach, but it requires participants to take PABA with meals and expensive laboratory procedures [12]. Creatinine-based criteria, such as 24-h urinary creatinine excretion, alone, or with expected urinary creatinine excretion, are often used but can be affected by age, body mass, protein intake, hydration status, and kidney function [11,12,13]. Comprehensive instruction and having investigators supervise and document the time of the first and last void of 24-h urine collection, and the report of any missed void may be helpful but requires substantial investigator time. 

The accuracy and usefulness of creatinine-based criteria to evaluate the completeness of 24-h urine collection has been systematically reviewed [11,12]. However, to our knowledge, there are no studies examining how different methods of assessing the completeness of 24-h urine collections influences the association of urine sodium excretion with health outcomes. We compared the mean sodium intake, and sodium intake and BP relationship from complete 24-h urine samples evaluated by creatinine-based criteria and participants’ self-reports using compiled data from Bangladesh.

## 2. Materials and Methods 

### 2.1. Data Sources 

We compiled data of three cohort studies from Bangladesh, carried out by the International Centre for Diarrhoeal Disease Research, Bangladesh (ICDDR,B). The study protocols were reviewed and approved by the Ethical Review Committee of ICDDR,B (PR-15096 and PR-18004). Informed written consent was obtained from all participants. The studies were predominantly implemented in southwest coastal Bangladesh where communities are exposed to brackish drinking water [14]. In total, we pooled 10,031 person-visits data. Several epidemiological studies from southwest coastal Bangladesh reported sodium intake through drinking water [15,16,17,18]. Nevertheless, increased sodium intake through drinking water is seasonal [19,20]. In the dry season, communities have limited or no drinking water sources other than brackish groundwater [19]. In contrast, water salinity is not significant during the wet season. Therefore, participants’ sodium intake from drinking water was low or minimum during the wet season. The first cohort study followed up 383 participants for two visits (742 person-visits) from southwest coastal Bangladesh during the wet season [21]. The second study was a population-based stepped-wedge randomized controlled trial in southwest coastal Bangladesh that followed up 1190 participants for five monthly visits (5745 person-visits) during the dry season [22,23]. The third cohort study followed up 293 participants from southwest coastal and 277 from non-coastal central Bangladesh for seven visits covering both wet and dry seasons (3547 person-visits data: 1773 from coastal and 1774 from non-coastal regions) [24]. 

### 2.2. Blood Pressure Measurement

We used a validated Omron^®^ HEM–907 (Kyoto, Japan) digital monitor [19] and an appropriate-sized cuff to measure the BP of the participants between 7 a.m. and 2 p.m. at all visits. Participants did not consume caffeine (e.g., tea, coffee) or food, and did not smoke or perform heavy physical activities 30 min before BP measurement. Participants rested for at least 5 min in a chair with back support in the sitting position with the arm supported at heart level. BP was measured three times and the mean BP was used for analyses. 

### 2.3. Demographic, Cardiovascular Risk Factor, and Physical Measures Data 

We collected participants’ demographics (age, sex, and religion) information, and their self-reported data on smoking, work-related physical activities, alcohol consumption, and sleep hours during all visits. Participants’ weight was measured in all visits, but height was measured once. To calculate households’ wealth, we collected information on households’ ownership of refrigerators, televisions, mobile phones, motorcycles, bicycles, sewing machines, chairs, tables, wristwatches, wardrobe wooden cots, motor pumps, rice husking machines, motorized rickshaw, cars, and access to electricity [25].

### 2.4. 24-h Urinary Sodium and Creatinine 

We collected participants’ 24-h urine at all visits. Each participant received a 4-L container for urine collection, and a small container to transfer the voided urine to the 4-L container. Participants were instructed not to collect their 1st-morning urine and to start urine collection by collecting and transferring the 2nd-morning void, and all voids up to and including the next morning’s 1st void. In the second and third cohort studies, participants or a family member who could read or write received a small questionnaire to record the times of the first and last void, and to report any missed voids. Field research assistants recorded the collected 24-h urine volume and then obtained a 15-mL urine sample from the 4-L container after stirring. They transported the urine samples to a field laboratory at 2–8 °C. Direct ion-selective electrode (ISE) methods were used to measure the urinary sodium with a semi-auto electrolyte analyzer (Biolyte2000, Bio-care Corporation, Taiwan, coefficient of variation: +5%). Creatinine was measured by the Jaffe reaction [26]. 

### 2.5. Methods of Evaluating the Completeness of 24-h Urine Collections 

We determined the completeness of 24-h urine collection based on self-reported information and creatinine-based methods. Two self-reported methods were considered: (1) Participants reported no missed voids, and (2) a 22–26 h interval between the time of the first and last urine void plus no missed voids. For the 2nd self-reported method, we standardized the urine volume for 24 h. We considered four creatinine-based methods: creatinine index ≥0.7 [12], measured 24-h urine creatinine (mCER) within 15% of Kawasaki predicted daily urine creatinine excretion [27], mCER within 25% of Kawasaki predicted daily urine creatinine excretion, and sex-specific mCER ranges (mCER 15–25 mg/kg/24 h for men and 10–20 mg/kg/24 h for women) [12,28]. Person-visits that reported any self-reported missed voids or incomplete urine collections were excluded from all creatinine-based methods. The creatinine index was defined as measured 24-h urine creatinine in mg/(21 × bodyweight) for women and measured 24-h urine creatinine in mg/(24 × bodyweight) for men [13]. The Kawasaki predicted daily urine creatinine excretion was calculated using the Kawasaki formula: −12.63 × age + 15.12 × weight + 7.39 × height − 79.90 for men, and −4.72 × age + 8.58 × weight + 5.09 × height − 74.50 for women [29].

### 2.6. Statistical Analyses

We derived the household wealth score by the principal component analysis using the household asset ownership data [24]. We then categorized the wealth score into wealth quintiles. We reported the mean and standard deviation or median and interquartile range, where appropriate, for continuous variables. We reported the proportion of the categorical covariates. We used the Kernel density plots to compare the distributions of urine sodium excretion by the different methods, and the two-sample t-test for evaluating the difference in mean urine sodium excretions across the methods. 

In a post hoc analysis, using data from all methods of assessing complete 24-h urine samples, we plotted the restricted cubic spline [30] to examine the trajectories of systolic and diastolic BP with increasing levels of urinary sodium excretion. The restricted cubic spline plots were initially created for men’s and women’s person-visits combined, and then separately. We used default four knots at 5th, 35th, 65th, and 95th percentiles to create the restricted cubic spline plots [31], after running the multilevel linear models with the participant-, household-, and community-level random intercepts to account for the clustering of data at different levels. We estimated the models using the maximum likelihood approach and clustered robust standard errors. We adjusted the models for age, sex, and body mass index (BMI), smoking and alcohol consumption, physical activity, religion, hours of sleep, and household wealth. Religion was considered as a covariate since the dietary habits across the two prominent religions are different in our study areas: Hindus are often vegetarian, but Muslims consume meat [32]. Sleep duration was considered a covariate because it can influence BP [33]. We did the Winsorization of the urine sodium at 0.2th and 99.8th percentiles to avoid distortion of the restricted cubic splines by the extreme urine sodium values at both tails of the distribution [34]. Winsorization replaced all values before the 0.2th percentile and after the 99.8th percentiles with the specific values of those percentiles. 

We also modeled urinary sodium as a categorical variable using tertiles of urinary sodium. Similar multilevel linear models to those described above were used to determine the associations of quartile 2, and 3 urinary sodium with BP compared to quartile 1. We reported the *p*-value for the linear trend determined by the orthogonal polynomials [35]. Since we identified a linear trend for the urine sodium–SBP relationship in tertile analyses, we also assessed the difference in mean blood pressure associated with 100-mmol increases in 24-h urine sodium (continuous exposure) using the similar multilevel linear regression described above. For the tertile and continuous urine sodium analyses, we reported the findings of unadjusted models; models adjusted for age, sex, and body mass index (BMI); and models additionally adjusted for smoking and alcohol consumption, physical activity, religion, hours of sleep, and household wealth. All analyses were conducted in STATA version 16.0. 

## 3. Results

Of the total of 9804 person-visits with 24-h urine collections, 7176 (73%) reported no missed voids, and 5524 (56%) reported 22–26 h of urine collection and no missed voids (Table 1). The creatinine index ≥0.7 included 5995 (61%) person-visits, the mCER within 15% of Kawasaki predicted creatinine included 1569 (16%) person-visits, the mCER within 25% of Kawasaki predicted creatinine included 2439 (25%) person-visits, and the sex-specific and mCER range included 2109 (22%) person-visits (Table 1). 

In total, 96% of the samples selected by mCER within 15% of Kawasaki predicted urine creatinine, and 94% of samples selected by mCER within 25% of Kawasaki predicted urine creatinine were also included by creatinine index ≥0.7. All (100%) of the samples selected by mCER within 15% of Kawasaki predicted urine creatinine were also selected by mCER within 25% of Kawasaki predicted urine creatinine. In total, 73% of the samples included by the sex-specific range were included by creatinine index ≥0.7, and 67% included by mCER within 15% of Kawasaki predicted urine creatinine were included by the sex-specific mCER range.

The mean age of the person-visits was the highest (44.3 ± 14.6 years) for the sex-specific mCER ranges, and the lowest (42.3 ± 14.2 years) for the creatinine index ≥0.7 criteria (Table 1). The proportion of male person-visits included in the complete 24-h urine collection subsample was 30% for both Kawasaki formula-based methods, 31% for the sex-specific mCER ranges, and 39% for all person-visits. The mean BMI was the highest for the sex-specific mCER ranges (23.3 ± 4.1 kg/m^2^) and the lowest (22.4 ± 4.1 kg/m^2^) for the creatinine index ≥0.7 criteria (Table 1).

The sex-specific mCER ranges had the lowest mean urine sodium excretion (137 (±66) mmol/24-h) and the creatinine index ≥0.7 method had the highest mean urine sodium excretion (169 (±74) mmol/24-h). Compared to the 158 (± 76) mmol/24-h mean urine sodium of the self-reported no missed urine (the reference method), all methods had a statistically different mean measured 24-h urine sodium (*p* < at the significance level of 0.05) (Table 1). The Kernel density plots suggest sex-specific mCER ranges had a lower mean and less dispersion of urine sodium excretion than the other methods, but all other methods had a similar dispersion of urinary sodium excretion (Figure 1). Women had higher mean 24-h urine sodium excretions than men (Table 1) in samples included by self-reported no missed urine and sex-specific mCER ranges (Table 1). Men consistently had higher 24-h urine creatinine concentrations than women. The sex-specific 24-h urine creatinine concentrations did not vary across methods except for the sex-specific mCER ranges, which included samples with lower 24-h urine creatinine concentrations for both sexes (Table 2). 

For all methods, the covariates-adjusted restricted cubic spline plots illustrated an initial steep positive sodium–SBP and sodium–DBP relationship (Figure 2). Afterward, we found a less steep positive sodium–SBP relationship for self-reported no missed voids, 22–26 h collection with no missed voids, creatinine index ≥0.7, and sex-specific mCER ranges; but a plateaued sodium–SBP relationship for the two Kawasaki formula-based methods (Figure 2). In contrast, a negative sodium–DBP relationship was observed after the initial steep positive relationship for the self-reported no missed voids, 22–26 h collection with no missed voids, and the two Kawasaki formula-based methods; and a plateaued sodium–DBP relationship for the creatinine index ≥0.7 and sex-specific mCER range methods (Figure 2). The sex-specific covariates-adjusted restricted cubic spline plots suggested a consistent positive sodium–SBP relationship across all methods for women but a varied sodium–SBP relationship for men (Figure 3).

In the fully adjusted model using tertiles of 24-h urinary sodium, there was a graded increase in BP with increases in sodium. Compared to the tertile 1 person-visits, those in tertile 3 had 3.36 (95% CI: 1.75, 4.96) mmHg higher SBP for the sex-specific mCER ranges, 2.97 (95% CI: 1.48, 4.46) mmHg higher SBP for the mCER within 15% of Kawasaki predicted creatinine, and 1.55 (95% CI: 0.37, 2.72) mmHg higher SBP for the mCER within 25% of Kawasaki predicted creatinine (Table 3). In the full-adjusted model, compared to the tertile 1 person-visits, those in tertile 3 had 1.66 (95% CI: 0.66, 2.66) mmHg higher DBP for the sex-specific mCER ranges, 1.20 (95% CI: 0.25, 2.15) mmHg higher DBP for the mCER within 15% of Kawasaki predicted creatinine, and 0.41 (95% CI: 0.01, 0.81) mmHg higher DBP for the creatinine index ≥0.7 method. A linear trend of the sodium–SBP relationship was found for all methods, but a linear trend of the sodium–DBP relationship was found only for the sex-specific mCER range, mCER within 15% of Kawasaki predicted creatinine, and creatinine index ≥0.7 methods (Table 3).

In the full-adjusted model, each 100 mmol/24-h increase in urinary sodium was associated with 0.68 (95% CI: 0.27, 1.08) mmHg higher SBP and 0.09 (95% CI: −0.17, 0.36) mmHg higher DBP for the creatinine index ≥0.7 method, 0.98 (95% CI: −0.07, 2.02) mmHg change in SBP and 0.30 (95% CI: −0.09, 0.70) mmHg higher DBP for mCER within 15% Kawasaki predicted urine creatinine, 0.87 (95% CI: 0.07, 1.67) mmHg higher SBP and 0.29 (95% CI: −0.12, 0.71) mmHg higher DBP for mCER within 25% Kawasaki predicted urine creatinine, and 1.96 (95% CI: 0.93, 2.99) mmHg higher SBP and 0.88 (95% CI: 0.16, 1.59) mmHg higher DBP for the sex-specific mCER range (Table 4).

## 4. Discussion

Twenty-four-hour urine collection is considered the most accurate method to assess dietary sodium intake. However, incomplete collection of urine samples is very common, particularly in population-based studies. Researchers across the world have used various methods to exclude incomplete 24-h urine samples. Our study demonstrates that the use of different methods led to significant differences in mean 24-h urine sodium excretion and altered the association of excreted sodium with BP, including the shape, magnitude, and significance levels. These differences were observed in all modeling strategies we assessed. Furthermore, we found a sex-specific difference in the urine sodium excretion and BP relationship. The different methods of assessing incomplete 24-h urine samples resulted in the exclusion of different proportions of urine collections. Additionally, the different methods resulted in differences in which specific collections were included or excluded. Likely, the indirect methods we evaluated may not effectively exclude all of the true incomplete 24-h urine samples and, at the same time, some complete 24-h urine samples may have been excluded in this Bangladeshi population. Since we did not have the gold standard PABA method, we are unable to determine which method performed the best.

The creatinine index ≥0.7 excludes samples with low urine creatinine, which is analogous to the exclusion of samples based on urine creatinine below 15 mg/kg/24-h [11]. The other three creatinine-based methods exclude samples with both low or high urine creatinine levels; therefore, they were more restrictive methods that included fewer urine samples. Of the creatinine-based methods, the trio of creatinine index ≥0.7, mCER within 25%, and mCER within 15% of Kawasaki predicted urine creatinine had some consistencies in terms of selecting complete 24-h urine samples because the latter method includes a sub-sample of urine collections selected by the former method. Therefore, mCER within 25% of Kawasaki predicted urine creatinine was a more restrictive approach than the creatinine index ≥0.7, and a further restrictive method was the mCER within 15% of Kawasaki predicted urine creatinine. Nevertheless, the sex-specific mCER ranges had considerable discrepancies with the other three creatinine-based methods. Nearly one in four samples included by the sex-specific ranges were excluded by creatinine index ≥0.7, and one in three samples included by mCER within 15% of Kawasaki predicted urine creatinine were excluded by the sex-specific mCER range. Such discrepancy raises concerns about the effectiveness of creatinine-based methods to evaluate the completeness of 24-h urine samples among the Bangladeshi population. 

Creatinine-based methods exclude samples if measured urine creatinine excretion levels are outside of the sex-specific physiological ranges; however, these methods consider neither participants’ age nor muscle mass, protein intake, and other variables influencing measured urine creatinine excretion. These methods were developed from a small number of participants; therefore, they may not reflect population-level characteristics and variability. For example, the creatinine index ≥0.7 was evaluated in 541 participants, the Kawasaki equation of completeness evaluation was developed from 487 participants, and the sex-specific mCER ranges was evaluated in 104 participants [11,12,29]. Significant variability of urinary creatinine excretion was observed in National Health and Nutrition Examination Survey participants across age groups, with young adults showing greater variation [36]. Therefore, creatinine-based methods may not be sensitive for a large population where participants are recruited from a wide age range. A systematic review suggested that the creatinine index ≥0.7 had a sensitivity of 49% and specificity of 88% when compared with the gold standard direct para-aminobenzoic acid excretion [12,37]. That means one in two samples selected by creatinine index ≥0.7 were not a complete 24-h urine sample. The sensitivity and specificity of the creatinine-based methods are likely different in our study population and that might have resulted in inaccurate inclusion of many incomplete 24-h urine samples in our analyses. Erroneous inclusion or exclusion of 24-h urine samples may explain the difference in the shape and magnitude of the urine sodium excretion and blood pressure relationship. 

We identified a plateaued relationship between the urine sodium and systolic blood pressure relationship above the 50th percentile distributions of urine sodium. Randomized controlled trials, in contrast, show a mostly linear relationship between dietary sodium intake and blood pressure, with potentially a steeper relationship at sodium intake below100 mmol/day. Residual confounding, with physical activity and other variables that co-relate with blood pressure and sodium intake may explain some of the plateauing. One of the other possibilities includes the presence of blood pressure-lowering minerals (e.g., potassium, magnesium) in the brackish water. In other analyses reported elsewhere [14], we identified that drinking brackish water is associated with not only a high intake of sodium but also higher intakes of magnesium, calcium, and potassium. We identified that urinary magnesium and calcium were associated with lower blood pressure, whereas urine sodium was associated with higher blood pressure. Further sex-specific plots identified such a plateaued or declined urine sodium and systolic blood pressure relationship above the 50th percentile distribution of urine sodium was in men, while women consistently had a positive upward relationship. The sex-specific differences can be attributable to factors such as sex-specific variation in the sodium intake and blood pressure association [38,39,40] and the sex-specific difference in muscle mass and urinary creatinine excretion. Creatinine is derived from skeletal muscle at a constant rate and excreted by the glomerulus. Creatinine is neither reabsorbed nor metabolized by the kidney tubular cells. South Asians have low muscle mass [40]; hence, creatinine-based methods may need adjustments for south Asians with low muscle mass. Males have a higher variation of urinary creatinine than females [41]; therefore, more urine samples of males are likely to be excluded by the creatinine-based indirect methods of 24-h completeness evaluation. In South Asian culture, males may have more variability in the collection of 24-h urine sample protocol since they work outside compared to females who stay at home. This could have led to the inclusion of a relatively lower proportion of samples of male person-visits across methods of complete 24-h sample determination. 

The sex-specific mCER ranges method excluded samples when measured urinary creatinine was low or high. The mean urine sodium excretion for the sex-specific mCER ranges was the lowest of all the methods, suggesting sex-specific mCER ranges included more data-points with low urine sodium excretion. Since the urine sodium and blood pressure relationship plots had a steep slope in the early segment of the plots, it also explains a higher magnitude of urine sodium and blood pressure association for sex-specific mCER ranges in linear analyses.

The study population had high sodium intake, compared to the WHO-recommended daily intakes of less than 2000 mg/day or 87 mmol/day. Only 8% of person-visits had urine sodium within 1500 mg/day or 65.2 mmol/day, the recommended daily intake by the Dietary Approaches to Stop Hypertension (DASH) study. We consistently found a steep slope of the urine sodium and blood pressure relationship curve in the spline plots’ initial sections. This suggests reducing daily sodium intake below the recommended daily intakes may reduce the population’s mean blood pressure markedly, albeit a small portion of person visits were within the recommended urine sodium excretion.

Our study has several important strengths and limitations. We had cross-sectional sodium exposure and blood pressure outcome data for several visits for the same individual. Many hemodynamic changes associated with salt load occur within 24 h. Therefore, blood pressure, along with sodium intake, measurement is particularly advantageous instead of a time lag between sodium intake and blood pressure measurement. We had a relatively large sample size that provided enough power for analyses, even after excluding samples by more restrictive methods of complete 24-h collection evaluation. Inclusion of participant-, household-, and community-level random intercepts largely captured time non-variant unmeasured confounders at each level and therefore minimized the bias. The use of Omron^®^ HEM–907 rather than using different instruments would have provided standardized blood pressure measurements, reducing the measurement errors in blood pressure measurement. Twenty-four-hour urine collection studies are recommended to incorporate estimation of the completeness of 24-h urine in a subsample using para-aminobenzoic acid [10], a gold standard approach to determining completeness. Lacking that component, we were unable to compare the creatinine-based and self-reported methods of completeness evaluation with para-aminobenzoic acid. Hence, we are not sure which urine samples are complete and which are not. Weather variables, particularly ambient temperature, rainfall, and humidity, affect blood pressure by influencing the peripheral resistance of skin vasculature [42,43,44]. We did not measure these weather variables, which precludes us from controlling for them. Nevertheless, future analyses need to be implemented after retrieving the weather data from the local weather station to see how weather variables influence the urine sodium and blood pressure relationship. Temperature and physical activities also influence the rate of sweating and sodium excretion [45], however, the sodium excretion through sweat decreases markedly among acclimatized individuals within weeks [46,47]. High sweat flow may overwhelm the ability to reabsorb sodium from sweat. Therefore, sweating can potentially lead to decreased urinary sodium excretion, particularly for men, who may not have been fully acclimatized during the study [23]. Hence, bias due to sodium excretion in sweat may also have influenced our analyses. Our study was limited to the Bangladeshi population predominantly from the southwest coastal region and may not be generalizable to other populations. Blood pressure has a diurnal variation, with morning blood pressure usually being higher than that measured around noon or afternoon [48]. We were unable to control for diurnal variation of blood pressure since we did not collect the exact time of blood pressure measurement, which may have introduced some bias in our estimates. 

## 5. Conclusions

Our findings suggest that the results of epidemiological studies evaluating the health effects of sodium intake can be influenced by different methods of evaluating the completeness of 24-h urine samples. It is therefore possible that other studies using 24-h urine collections without considering the completeness of urine samples may also have provided altered and unreliable associations between sodium and health outcomes. Low-quality epidemiological studies lead to more debates causing controversy and could impact public health policy on sodium reduction. It is vital to ensure the 24-h urine collections are complete in epidemiological studies, preferably using para-aminobenzoic acid to assess the completeness of the collected urine samples, so as to provide robust evidence on sodium intake and health outcomes. Further research on the evaluation of indirect methods of complete 24-h urine samples is needed, particularly to compare them with para-aminobenzoic acid excretion. 

## Figures and Tables

**Figure 1 nutrients-12-02772-f001:**
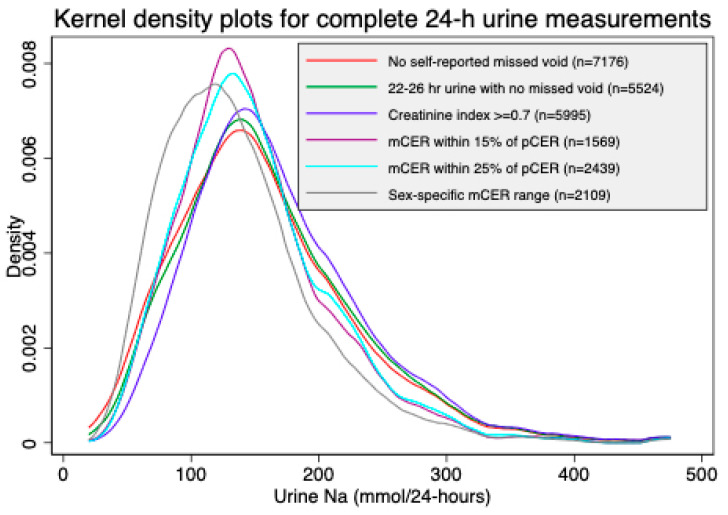
Kernel density plots of measured 24-h urine sodium excretion based on different methods of calculating complete 24-h urine samples. mCER: measured 24-h urine creatinine. excretion; pCER: predicted 24-h urine creatinine excretion.

**Figure 2 nutrients-12-02772-f002:**
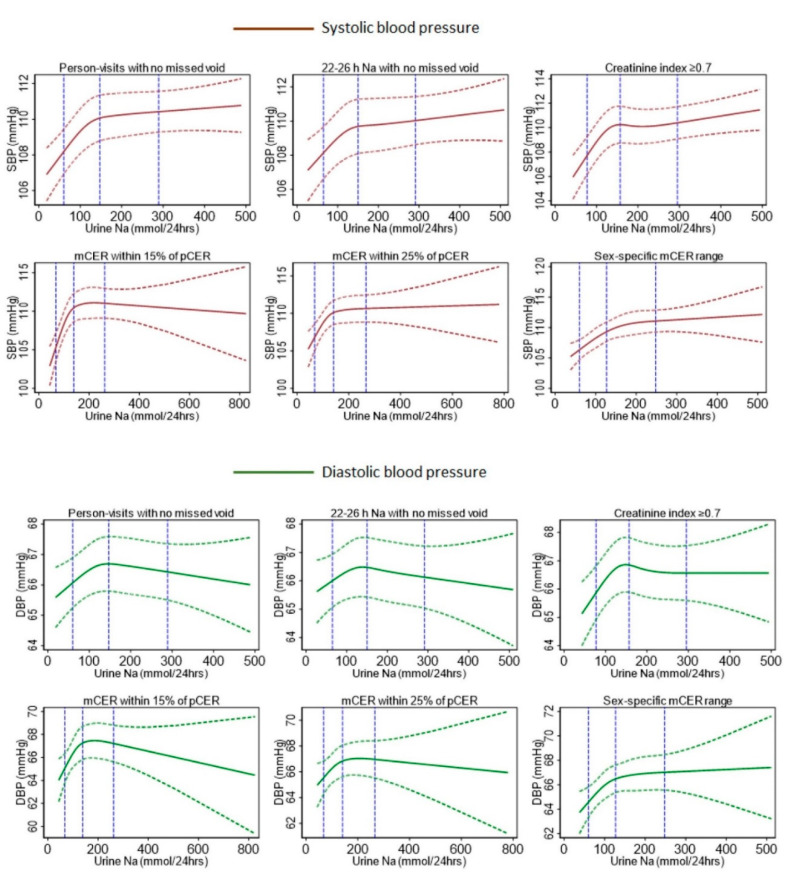
Restricted cubic spline pots of urinary sodium and blood pressure relationship from different method of evaluating the completeness of 24-h urine collection. Models were adjusted for age, sex, BMI, smoking status, physical activity, sleep, alcohol consumption, religion, and household wealth. Vertical dotted blue dotted lines indicate the 5th, 50th, and 95th percentile distribution of urine sodium excretion. mCER: measured 24-h urine creatinine excretion; pCER: predicted 24-h urine creatinine excretion.

**Figure 3 nutrients-12-02772-f003:**
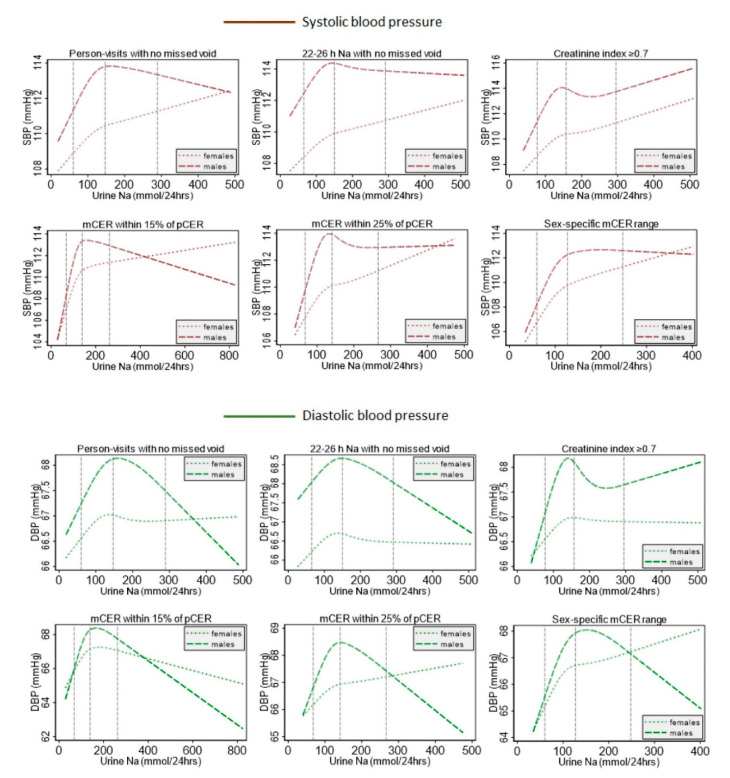
Restricted cubic spline plots of the sex-specific relationship between urinary sodium and blood pressure from different methods of evaluating the completeness of 24-h urine collection. Models were adjusted for age, sex, BMI, smoking status, physical activity, sleep, alcohol consumption, religion, and household wealth. Vertical dotted blue dotted lines indicate the 5th, 50th, and 95th percentile distribution of urine sodium excretion. mCER: measured 24-h urine creatinine excretion; pCER: predicted 24-h urine creatinine excretion.

**Table 1 nutrients-12-02772-t001:** Age, sex, and BMI distributions of person-visits in subsamples of the study population captured by different methods of evaluating the completeness of 24-h urine samples.

Characteristics	All Person-Visits of 24-h Urine Collected	Self-Reported No Missed Voids	22–26 h Urine Samples with No Missed Voids	Creatinine Index ≥0.7 & No Missed Voids	mCER within 15% of Kawasaki pCER and No Missed Voids	mCER within 25% of Kawasaki pCER and No Missed Voids	mCER 15–25 mg/kg/24 h (Men) and 10–20 mg/kg/24 h (Women), and No Missed Voids
Person-visits included, N (%)	9804 (100%)	7176 (73%)	5524 (56%)	5995 (61%)	1569 (16%)	2439 (25%)	2109 (22%)
Age, mean (SD)	42.7 (14.4)	42.8 (14.5)	42.8 (14.4)	42.3 (14.2)	42.9 (14.6)	43.0 (14.5)	44.3 (14.6)
Age category, % (n)							
20- < 30 years	20 (1974)	20 (1490)	20 (1110)	21 (1300)	18 (295)	18 (457)	16 (342)
30- < 40 years	29 (2829)	28 (2093)	28 (1560)	29 (1777)	30 (484)	30 (731)	29 (605)
40- < 50 years	20 (1937)	20 (1468)	20 (1109)	20 (1270)	18 (286)	18 (456)	19 (392)
50- < 60 years	16 (1617)	16 (1224)	17 (909)	16 (1003)	15 (249)	16 (391)	19 (391)
60- < 70 years	10 (1006)	11 (785)	11 (604)	10 (630)	12 (186)	12 (303)	13 (274)
≥70 years	5 (441)	5 (372)	4 (239)	4 (275)	7 (108)	6 (140)	5 (105)
Male sex, % (n)	39 (3802)	36 (2677)	34 (1893)	35 (2198)	30 (473)	30 (742)	31 (664)
BMI, mean (SD)	22.4 (3.9)	22.5 (4.1)	22.6 (3.8)	22.4 (4.1)	22.9 (4.0)	22.9 (4.0)	23.3 (4.1)
BMI categories, %							
Underweight	15 (1445)	14 (1070)	14 (748)	15 (910)	14 (218)	13 (331)	11 (227)
Normal weight	43 (4251)	43 (3169)	42 (2309)	43 (2674)	38 (609)	39 (963)	39 (821)
Overweight	31 (3081)	32 (2348)	33 (1852)	32 (1995)	34 (550)	34 (846)	35 (740)
Obese	11 (1027)	11 (854)	11 (622)	11 (676)	14 (231)	14 (338)	15 (321)
24-h urine Na for both sexes, mean(SD) [*p*-value ^¥^]	155 (76) [0.011]	158 (76) [reference]	162 (76) [0.003]	169 (74) [<0.001]	151 (79) [<0.001]	152 (74) [<0.001]	137 (66) [<0.001]
24-h urine Na for males, mean(SD) [*p*-value ^¥^]	149 (79) [0.003]	155 (81) [reference]	161 (84) [0.016]	169 (80) [<0.001]	151 (99) [0.344]	151 (87) [0.246]	133 (57) [<0.001]
24-h urine Na for females, mean(SD) [*p*-value ^¥^]	158 (74) [0.1646]	160 (73) [reference]	163 (72) [0.062]	169 (71) [<0.001]	151 (68) [<0.001]	153 (68) [<0.001]	139 (70) [<0.001]

BMI: body mass index; mCER: measured 24-h urine creatinine excretion; pCER: predicted 24-h urine creatinine excretion. ^¥^: *p*-value for mean difference determined by two sample *t*-test.

**Table 2 nutrients-12-02772-t002:** Sex-specific 24-h urine volume and 24-h urine creatinine concentrations (mg/dL) from different methods of evaluating the completeness of 24-h urine samples.

Methods of Evaluating 24-h Urine Samples	Urine Volume in L, Median (IQR)		Urine Creatinine Concentration in mg/dL, Median (IQR)	
	Men	Women	Men	Women
No self-reported missed voids	1.93 (1.37–2.65)	1.92 (1.41–2.57)	65 (48–92)	53 (39–72)
22–26 h urine samples with no missed voids	1.95 (1.40–2.69)	1.95 (1.45–2.60)	66 (49–92)	53 (39–72)
Creatinine index ≥0.7 and no missed voids	2.09 (1.55–2.81)	2.03 (1.54–2.67)	68 (51–96)	54 (40–75)
mCER within 15% of Kawasaki pCER and no missed voids	1.87 (1.32, 2.63)	1.83 (1.36–2.47)	68 (52–94)	52 (39–68)
mCER within 25% of Kawasaki pCER and no missed voids	1.80 (1.32–2.53)	1.85 (1.35–2.49)	67 (52–95)	52 (39–69)
mCER 15–25 mg/kg/24 h (men) and 10–20 mg/kg/24 h (women), and no missed voids	1.74 (1.24–2.33)	1.65 (1.23–2.18)	62 (49–85)	50 (39–65)

mCER: measured 24-h urine creatinine excretion; pCER: predicted 24-h urine creatinine excretion.

**Table 3 nutrients-12-02772-t003:** Association between the tertile of urinary sodium and blood pressure.

Methods for Complete 24-h Urine Assessment	Systolic BP (mmHg)				Diastolic BP (mmHg)			
	Tertile 1	Tertile 2	Tertile 3	*p*-Value for Trend	Tertile 1	Tertile 2	Tertile 3	*p*-Value for Trend
No self-reported missed voids								
Model 1	Ref	0.60 (0.01, 1.19)	0.98 (0.36, 1.60)	0.002	Ref	−0.04 (−0.42, 0.35)	0.14 (−0.24, 0.52)	0.463
Model 2	Ref	0.75 (0.15, 1.35)	1.14 (0.57, 1.71)	<0.001	Ref	0.01 (−0.34, 0.36)	0.12 (−0.26, 0.49)	0.545
Model 3	Ref	0.73 (0.14, 1.33)	1.15 (0.58, 1.72)	<0.001	Ref	−0.01 (−0.35, 0.33)	0.13 (−0.25, 0.50)	0.508
22–26 h urine samples with no missed voids								
Model 1	Ref	0.62 (0.06, 1.17)	1.03 (0.35, 1.70)	0.003	Ref	0.01 (−0.48, 0.48)	0.12 (−0.37, 0.61)	0.625
Model 2	Ref	0.75 (0.22, 1.27)	1.09 (0.44, 1.73)	0.001	Ref	0.03 (−0.39, 0.45)	0.01 (−0.46, 0.47)	0.984
Model 3	Ref	0.74 (0.22, 1.26)	1.10 (0.47, 1.73)	<0.001	Ref	0.02 (−0.39, 0.43)	0.01 (−0.44, 0.46)	0.963
Creatinine index ≥0.7 & no missed voids								
Model 1	Ref	1.13 (0.55, 1.72)	1.31 (0.71, 1.92)	<0.001	Ref	0.51 (0.12, 0.89)	0.53 (0.18, 0.88)	0.003
Model 2	Ref	1.24 (0.65, 1.84)	1.33 (0.75, 1.91)	<0.001	Ref	0.52 (0.15, 0.90)	0.41 (0.02, 0.80)	0.037
Model 3	Ref	1.21 (0.62, 1.81)	1.33 (0.74, 1.92)	<0.001	Ref	0.50 (0.12, 0.87)	0.41 (0.01, 0.81)	0.047
mCER within 15% of Kawasaki predicted daily creatinine								
Model 1	Ref	2.21 (0.82, 3.60)	2.56 (1.03, 4.09)	0.001	Ref	0.95 (−0.01, 1.91)	1.38 (0.41, 2.35)	0.005
Model 2	Ref	2.49 (1.02, 3.95)	2.96 (1.45, 4.46)	0.001	Ref	0.84 (−0.13, 1.81)	1.88 (0.24, 2.14)	0.014
Model 3	Ref	2.49 (1.02, 3.97)	2.97 (1.48, 4.46)	<0.001	Ref	0.87 (−0.11, 1.84)	1.20 (0.25, 2.15)	0.013
mCER within 25% of Kawasaki predicted daily creatinine								
Model 1	Ref	0.63 (−0.44, 1.69)	1.41 (0.17, 2.64)	0.025	Ref	0.18 (−0.51, 0.88)	0.72 (−0.12, 1.56)	0.095
Model 2	Ref	0.80 (−0.28, 1.88)	1.54 (0.34, 2.74)	0.012	Ref	0.08 (−0.63, 0.80)	0.46 (−0.40, 1.31)	0.295
Model 3	Ref	0.79 (−0.28, 1.85)	1.55 (0.37, 2.72)	0.010	Ref	0.08 (−0.65, 0.80)	0.46 (−0.40, 1.31)	0.295
mCER 15–25 mg/kg/24 h for men and 10–20 mg/kg/24 h for women								
Model 1	Ref	2.31 (1.09, 3.52)	3.83 (2.19, 5.46)	<0.001	Ref	1.59 (0.81, 2.38)	2.29 (1.33, 3.25)	<0.001
Model 2	Ref	2.01 (0.88, 3.15)	3.34 (1.76, 4.91)	<0.001	Ref	1.23 (0.45, 2.02)	1.62 (0.63, 2.61)	0.001
Model 3	Ref	2.01 (0.85, 3.16)	3.36 (1.75, 4.96)	<0.001	Ref	1.23 (0.45, 2.02)	1.66 (0.66, 2.66)	0.001

No self-reported missed voids—urine sodium tertile 1: <121.9 mmol/day; tertile 2: >=121.9 to <175.8 mmol/day; tertile 3: >=175.8 mmol/day. 22–26 h urine samples with no missed voids —urine sodium tertile 1: <126.4 mmol/day; tertile 2: >=126.4 to <178.9 mmol/day; tertile 3: >=178.9 mmol/day. Creatinine index ≥0.7—urine sodium tertile 1: <133.7 mmol/day; tertile 2: >=133.7 to <186.1 mmol/day; tertile 3: >=186.1 mmol/day. mCER within 15% of Kawasaki predicted daily creatinine —urine sodium tertile 1: <119.4 mmol/day; tertile 2: >=119.4 to <162.9 mmol/day; tertile 3: >=162.9 mmol/day. mCER within 25% of Kawasaki predicted daily creatinine: <119.7 mmol/day; tertile 2: >=119.7 to <166.1 mmol/day; tertile 3: >=166.1 mmol/day. Sex-specific mCER ranges—urine sodium tertile 1: <104.2 mmol/day; tertile 2: >=104.2 to <151 mmol/day; tertile 3: >=151 mmol/day. mCER: measured 24-h urine creatinine excretion; pCER: predicted 24-h urine creatinine excretion. Model 1: Unadjusted; Model 2: adjusted for age, sex, and body mass index (BMI); Model 3: additionally, adjusted for physical activities, smoking status, alcohol consumption, sleep hours, religion, and household wealth. * *p*-value < 0.05 suggestive of a linear trend.

**Table 4 nutrients-12-02772-t004:** Difference in mean systolic BP (mmHg) for 100 mmol or 2300 mg per 24-h increase in 24-h urine sodium.

Methods for Complete 24-h Urine Assessment	Model 1 β (95% CI)	Model 2 β (95% CI)	Model 3 β (95% CI)
Systolic blood pressure			
No self-reported missed voids (*n* = 7176)	0.57 (0.28, 0.86)	0.64 (0.34, 0.94)	0.64 (0.34, 0.94)
22–26 h urine samples with no missed voids (n = 5524)	0.54 (0.21, 0.87)	0.53 (0.19, 0.87)	0.54 (0.21, 0.87)
Creatinine index ≥0.7 and no missed voids (n = 5995)	0.67 (0.29, 1.06)	0.68 (0.28, 1.09)	0.68 (0.27, 1.08)
mCER within 15% of Kawasaki pCER and no missed voids (n = 1569)	0.86 (−0.11, 1.84)	0.97 (−0.08, 2.02)	0.98 (−0.07, 2.02)
mCER within 25% of Kawasaki pCER and no missed voids (n = 2439)	0.81 (0.05, 1.58)	0.87 (0.06, 1.67)	0.87 (0.07, 1.67)
mCER 15–25 mg/kg/24 h (men) and 10–20 mg/kg/24 h (women), and no missed voids (n = 2109)	2.25 (1.10, 3.39)	1.94 (0.90, 2.98)	1.96 (0.93, 2.99)
Diastolic blood pressure			
No self-reported missed voids (n = 7176)	0.06 (−0.15, 0.27)	0.02 (−0.18, 0.23)	0.03 (−0.18, 0.24)
22–26 h urine samples with no missed voids (n = 5524)	0.04 (−0.22, 0.30)	−0.04 (−0.30, 0.21)	−0.04 (−0.30, 0.22)
Creatinine index ≥0.7 and no missed voids (n = 5995)	0.16 (−0.09, 0.40)	0.09 (−0.16, 0.35)	0.09 (−0.17, 0.36)
mCER within 15% of Kawasaki pCER and no missed voids (n = 1569)	0.38 (−0.07, 0.83)	0.29 (−0.1, 0.69)	0.30 (−0.09, 0.70)
mCER within 25% of Kawasaki pCER and no missed voids (n = 2439)	0.41 (−0.03, 0.84)	0.29 (−0.12, 0.71)	0.29 (−0.12, 0.71)
mCER 15–25 mg/kg/24 h (men) and 10–20 mg/kg/24 h (women), and no missed voids (n = 2109)	1.30 (0.53, 2.08)	0.85 (0.13, 1.56)	0.88 (0.16, 1.59)

mCER: measured 24-h urine creatinine excretion; pCER: predicted 24-h urine creatinine excretion. Model 1: Unadjusted; Model 2: adjusted for age, sex, and body mass index (BMI); Model 3: additionally, adjusted for physical activities, smoking status, alcohol consumption, sleep hours, religion, and household wealth.

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
