# Peer review of "Urinary Sodium Excretion and Blood Pressure Relationship across Methods of Evaluating the Completeness of 24-h Urine Collections"

_nutrients, 2020, doi:10.3390/nu12092772_

Round 1

Reviewer 1 Report

All changes have been made properly

Author Response

Thank you very much for agreeing with our responses. 

Sincerely,

Anu Mohd Naser

Reviewer 2 Report

The authors have responded to most of my suggestions. However, the point to reanalyze data using urinary Na/mg crea was declined. I still think that this approach could add to the quality of the manuscript.

Therefore, the authors should perform some analyses and then decide if this approach is beneficial or not. I would like to see those at least in the cover letter.

The ratio should be calculated as urinary Na conc in mM divided by the urinary creatinine concentration (mg/dL). Values are from 24 h urine samples.

Author Response

Thank you so much for your comments and suggestions. We have conducted additional analyses as per the suggestions of the reviewer. Please find our detailed responses attached.

Sincerely,

Abu Mohd Naser

Round 2

Reviewer 2 Report

Thank you for doing the interesting reanalysis which indeed is difficult to interpret and does not improve the data.

Therefore, I agree that they be not included.

This manuscript is a resubmission of an earlier submission. The following is a list of the peer review reports and author responses from that submission.

Round 1

Reviewer 1 Report

ENa, ECr and BP in Bangladesh.  

This is a very important study.  I list my suggestions below.  

  1. Abstract:  Suggest changing this sentence:  Studies examining 24-hour urine sodium in relation to health outcomes 38 will have different results based on how urine collections are complete urine samples are evaluated are deemed as complete.
  2. This study explores the relationship between a population’s BP, Na intake and ECr using a questionnaire, ECr analysis and the Kawsaki formula to find the “best” way to sort out those who are poor urine collectors for population studies of BP and Na intake.
  3. The introduction mentions high drinking water salt content as a possible factor in high ENa and BP in this population. Please add the seasonal variations in salt concentrations as almost all drinking water Na concentrations around the world I have seen are so low that one almost cannot drink enough water a day to affect ENa or affect BP much.  However the very high water Na levels reported from this are must be an exception.  Thus drinking 2 L of the highest salt well water would just get a person up to the DASH goal of 1,500. As this study shows once ENa exceeds 150 mM day there is almost no further increase in BP do to the plateau relationship of ENa to BP.
  4. If this plateau is real, and I suspect it is based on other studies then this may be the reason other studies have shown a poor relationship between ENa and events. I do not believe any studies have looked at events is a population with a plateau.  This should be discussed I would think.
  5. Have the investigators documented the accuracy/reliability of the questions they use for categorization/classification? If not, this should be mentioned in the discussion as a limitation.
  6. The body size, BMI of the patients from which the Kawasaki data was derived may be different from those from Bangladesh. Did the investigators consider doing a formula from the Bangladesh sample to estimate ECr based on age, weight, etc.  This might be more accurate in this population to do follow up studies.  I suspect they can come up with a formula based on their own data and test it against the other formulas. 
  7. Is the left end of the splint plots for ENa stop at the lowest value in the sample or do they extend past that. I would prefer that they only extend down to the lowest values found. I would suspect no one in the highest salt content water would have an ENa of <1,500 mg/d?
  8. A point should be made of what percent of their population with elevated BP has a recommended ENa of less than 1,500 mg/day to as recommended by current guidelines. My guess is very few if any. If they also measure EK this should be included in the analysis as a plot of BP and urine Na/K ratio would be most informative.  However, as the reported sodium content can go as high as 700 mg Na/L, they would just get enough to get up to DASH goal, 1500 mg Na/d if they consumed 2L of this water each day.  As figure 2 shows that level of ENa/24 hr was seen in less than 5% of the population.  However,  it is important to note that this is in the center of the steepest part of the Na/24 hr and BP relationship.   it would be difficult for anyone in those areas to get down to the DASH goal while drinking this water.  This should be in the discussion.
  9. The issue of lower ENa in men, as mentioned in the discussion, may be related to working outside of the home.  I assume few have air conditioners at home?  Even though a person has lowered their sweat Na to low values as part of their heat adaptation it is still possible to lose large amounts of sweat when working in the heat. See Ladell WSS, The changes in water and chloride distribution during heavy sweating. J. Physiology 108:440-450, 1948 and studies by Conn JW.  This high sweat flow “overwhelms” the ability of the sweat gland to reabsorb Na as the sweat is not residing in the sweat ducts long for Na to be reabsorbed, even in the most heat adapted.  This should be mentioned in their discussion.  Are there any reports of sweat volume/sweat Na losses under working conditions when heat adapted in these areas to support the conjecture that heat adaptation accounts for the lower ENa. 

  1. The local weather conditions and ENa. Even though they did not collect weather temps etc this should be available from the local weather data collection sites I would think. The sweat loss would be expected to higher and the ENa lower during the hottest times. At least at rest.  It would be useful to see it this data can be retrieved and examined for variations is ENa.
  2. NHANES studies (Cogswell et al) and this study report some of the strongest relationships between Na and BP published. While this may be related to urine collection issues they do not discuss the standardized BPs used in this study(Omron 907) and NHANES (mercury with highly trained observers using mercury manometers) as an additional source of the stronger relationship.  This should be discussed as possibly one of the reasons for the stronger relationship between ENa and BP. Both have been measure better than most other studies..
  3. The world wide recommendation for everyone with HTN is to move to the DASH eating plan. It seems of interest to see if populations are moving to this rather than just the lower Na intake.  Adding urine K would be of interest as the Na/K ratio is more closely related to BP in some studies.  Also, when one is monitoring compliance with a recommended DASH eating plan (recommend for everyone with HTN) the EK adds an additional check on compliance.  See: Potassium and Blood Pressure: How to test the effects of the DASH Diet in your patients with hypertension.  Hypertension Journal 2017;3:37-41.

  1. As noted by the Mars simulation study a single 24 hr urine is of little value in estimating a person’s intake even when the intake is fixed. The authors state they collected urine samples more than once in some participants and in one cohort for 7 visits.  What does that data show?  Is there any data on day to day variation in Na/K intake in these areas?

  1. It might be easier to use a sleep urine at least to detect those who are eating the lower Na intakes which should be the goal eventually. It is certainly easier to collect and better to see is a patient is on a lower Na intake.

  1. Fig 2 would be better if the 5% line was green (good), the 50% line is yellow (not so good) and the 95 as red (bad) for goal purposes. These plots should be very useful for teaching purposes for professionals and patients, i.e. the steep slope and the plateau.  Their plots show the BP-ENa relationship extends down to maybe 20 mM Na per 24 hours. Is this an extension for drawing the curve or did they actually have 24 hr urines that were this low? 

  1. If “weather” plays a role in ENa I would expect that ENa might be related to average daily temperature which could be assessed by using weather data during the study.

  1. Their cubic spline plots also demonstrate a dose effect with what looks like a plateau. If one tests the effect only for those with an ENa above the 50th percentile is it linear or non-linear? It does appear that in some plots above the 50th they actually show a lower BP at very high ENa.  Perhaps these folks are ill (have a lower BP) due to an inability to retain Na? 

Author Response

### Reviewer 1

  1. Abstract:  Suggest changing this sentence:  Studies examining 24-hour urine sodium in relation to health outcomes 38 will have different results based on how urine collections are complete urine samples are evaluated are deemed as complete.

Response: We have addressed reviewer’s suggestion in the last sentence of the abstract.

  1. This study explores the relationship between a population’s BP, Na intake and ECr using a questionnaire, ECr analysis and the Kawsaki formula to find the “best” way to sort out those who are poor urine collectors for population studies of BP and Na intake.

Response: Thanks. We agree with the reviewer that we implemented several published approaches of evaluating completeness of the 24-hour urine collection, including the Kawasaki formula. The objective was to assess how sodium intake and blood pressure relationship varies across different methods of evaluating complete 24-hour urine collection.

  1. The introduction mentions high drinking water salt content as a possible factor in high ENa and BP in this population. Please add the seasonal variations in salt concentrations as almost all drinking water Na concentrations around the world I have seen are so low that one almost cannot drink enough water a day to affect ENa or affect BP much.  However the very high water Na levels reported from this are must be an exception.  Thus drinking 2 L of the highest salt well water would just get a person up to the DASH goal of 1,500. As this study shows once ENa exceeds 150 mM day there is almost no further increase in BP do to the plateau relationship of ENa to BP.

Response: Thanks for the suggestions. We have updated the information on seasonal variation of water salinity in the “Data sources” section of the revised manuscript. We agree with the reviewer that southwest coastal Bangladesh is an unusual setting where people may have high sodium intake through drinking water due to complex multifactorial seawater intrusion. Many coastal communities across the world are experiencing such seawater intrusion due to sea-level rise. We have cited relevant epidemiological studies highlighting the high sodium intake in the study region. 

In response to the reviewer’s comment on plateaued urine sodium and BP relationship above 150 mmol/day, we believe there could be several explanations for our study population. We have included more extensive discussio in the revised manuscript.  

  1. If this plateau is real, and I suspect it is based on other studies then this may be the reason other studies have shown a poor relationship between ENa and events. I do not believe any studies have looked at events is a population with a plateau.  This should be discussed I would think.

Response: Thanks for raising this point. As we have responsed the previous comment of the reviewer, we have particularly emphasized explanation of plateaued relationship in the discussion section of revised manuscript. It is notable that RCT show a linear association of sodium intake and blood pressure which is now referenced in the discussion.

  1. Have the investigators documented the accuracy/reliability of the questions they use for categorization/classification? If not, this should be mentioned in the discussion as a limitation.

Responses: Thanks for the comment, but we are not very clear about this comment. We collected questionnaire information only for capturing cardiovascular health risk factor data such as smoking, physical activity, household assets, sleep hours, etc. The blood pressure, urine creatinine, and sodium measurements were objective, and the evaluation methods for the completeness of 24-hour urine samples were done based on published formulas and literature. There are published systematic reviews on the effectiveness of these approaches. We have clarified this in the introduction section of this manuscript.

  1. The body size, BMI of the patients from which the Kawasaki data was derived may be different from those from Bangladesh. Did the investigators consider doing a formula from the Bangladesh sample to estimate ECr based on age, weight, etc.  This might be more accurate in this population to do follow up studies.  I suspect they can come up with a formula based on their own data and test it against the other formulas. 

Response: Thanks for the the great suggestion. We agree with the reviewer that the demographics (age and sex), anthropometric (height and weight), and BMI data can be used to develop a formula for the Bangladeshi population to predict urinary creatinine. We’re currently considering the usefulness of the development of such formula, but at the same time, evaluating the pros and cons of using such formula for different purposes. Our big concern is that there are several other variables that may influence urinary creatinine.

  1. Is the left end of the splint plots for ENa stop at the lowest value in the sample or do they extend past that. I would prefer that they only extend down to the lowest values found. I would suspect no one in the highest salt content water would have an ENa of <1,500 mg/d?

Response: Thanks for the queries. The end of the spline plots will not extend beyond the values of the urine sodium or variable of the X-axis at either low or high end. In reponse to reviewer’s second part of the comment, we wanted to clarify that we compiled >10,000 person-visits data from three cohort studies. Participants did not always consume high salt containing water. In particular, during the wet season, participants’ salt intake from the drinking water was minimum. We have included this discussion in the revised manuscript.

  1. A point should be made of what percent of their population with elevated BP has a recommended ENa of less than 1,500 mg/day to as recommended by current guidelines. My guess is very few if any. If they also measure EK this should be included in the analysis as a plot of BP and urine Na/K ratio would be most informative.  However, as the reported sodium content can go as high as 700 mg Na/L, they would just get enough to get up to DASH goal, 1500 mg Na/d if they consumed 2L of this water each day.  As figure 2 shows that level of ENa/24 hr was seen in less than 5% of the population.  However,  it is important to note that this is in the center of the steepest part of the Na/24 hr and BP relationship.   it would be difficult for anyone in those areas to get down to the DASH goal while drinking this water.  This should be in the discussion.

Response: As ther reviewer suggested, we have included the proportion of person-visits that had 24-hour urine sodium below 1500 mg/day or 65 mmol/day. Only 8% of the person-visits had urine sodium intake below 1500 mg/day. We agree with the reviewer that in this study population, the average sodium intake was way high above the DASH goal. The reason is not solely due to sodium intake through drinking water, but also due to high use of salt to cook food. As suggested by the reviewer, we have included this discussion in the revised manuscript.

Concerning urine Na less than 5th percentile in Figure 2, we agree that these person-visits likely have sodium intake within the DASH recommended intake. However, we believe some of these data points may be also due to incomplete collection of 24-hour urine. Such data points decreased substantially for methods that excluded samples based on low predicted urine creatinine such as mCER within 15% of pCER, mCER within 25% of pCER, and sex-specific mCER ranges, as presented in Figure 2.

  1. The issue of lower ENa in men, as mentioned in the discussion, may be related to working outside of the home.  I assume few have air conditioners at home?  Even though a person has lowered their sweat Na to low values as part of their heat adaptation it is still possible to lose large amounts of sweat when working in the heat. See Ladell WSS, The changes in water and chloride distribution during heavy sweating. J. Physiology 108:440-450, 1948 and studies by Conn JW.  This high sweat flow “overwhelms” the ability of the sweat gland to reabsorb Na as the sweat is not residing in the sweat ducts long for Na to be reabsorbed, even in the most heat adapted.  This should be mentioned in their discussion.  Are there any reports of sweat volume/sweat Na losses under working conditions when heat adapted in these areas to support the conjecture that heat adaptation accounts for the lower ENa. 

Response: We agree with the reviewer that lower urine sodium in men could be linked to working outside of the home in the sun in a tropical climate. The reviewer is correct that only a few participants have air conditioners at home. We have included the possibility of sodium lost with sweat in the revised manuscript. Unfortunately, no published data is available for the loss of sodium through sweat for the Bangladeshi population.

  1. The local weather conditions and ENa. Even though they did not collect weather temps etc this should be available from the local weather data collection sites I would think. The sweat loss would be expected to higher and the ENa lower during the hottest times. At least at rest. It would be useful to see it this data can be retrieved and examined for variations is ENa.

Response: Thanks for the insight. Yes, the weather data can be retrieved from the local weather office. Nevertheless, such data is not publicly available. We agree with the reviewer that it will be great to retrieve that data and conduct further analyses to explain how weather variables are associated with sodium excretions and health outcomes in this setting.  Such an analysis is beyond the scope of this article.

  1. NHANES studies (Cogswell et al) and this study report some of the strongest relationships between Na and BP published. While this may be related to urine collection issues they do not discuss the standardized BPs used in this study(Omron 907) and NHANES (mercury with highly trained observers using mercury manometers) as an additional source of the stronger relationship.  This should be discussed as possibly one of the reasons for the stronger relationship between ENa and BP. Both have been measure better than most other studies.

Response: Thanks for the suggestions. We have discussed this strength of the study in the revised manuscrtipt.

  1. The world wide recommendation for everyone with HTN is to move to the DASH eating plan. It seems of interest to see if populations are moving to this rather than just the lower Na intake.  Adding urine K would be of interest as the Na/K ratio is more closely related to BP in some studies.  Also, when one is monitoring compliance with a recommended DASH eating plan (recommend for everyone with HTN) the EK adds an additional check on compliance.  See: Potassium and Blood Pressure: How to test the effects of the DASH Diet in your patients with hypertension.  Hypertension Journal 2017;3:37-41.

Response: We agree with the reviewer that including potassium in the analyses, or considering urine Na/K ratio would give additional insights. Nevertheless, we believe that considering the urinary Na/K ratio and blood pressure association can be thoroughly evaluated in another paper that we are considering. Inclusion of that additional analyses in the current manuscript will be too lengthy and may overwhelm the readers. 

  1. As noted by the Mars simulation study a single 24 hr urine is of little value in estimating a person’s intake even when the intake is fixed. The authors state they collected urine samples more than once in some participants and in one cohort for 7 visits.  What does that data show?  Is there any data on day to day variation in Na/K intake in these areas?

Response: Thanks for raising this point. We appreciate reviewer’s comment, but we did not report that result in this manuscript because we think this is not directly related with the objective of the research question we wanted to answer. Yes, we observed variation of participants’ 24-hour urine over different follow-up visits for the same individual. Usually, we observed less sodium intake during the wet season, when participants’ sodium intake through drinking water was negligible. Nevertheless, we did not look how the Na/K ratio changed across visits. We believe this will be an interesting research question to explore thoroughly in another paper.

  1. It might be easier to use a sleep urine at least to detect those who are eating the lower Na intakes which should be the goal eventually. It is certainly easier to collect and better to see is a patient is on a lower Na intake.

Response: We agree with the reviewer that a sleep urine test will provide valuable information about lower sodium intakes, which can be implemented in future studies in the region. In the three cohort studies used for this paper, we predominantly collected 24-hour urine samples. Unfortunately, we did not conduct any sleep urine test.

  1. Fig 2 would be better if the 5% line was green (good), the 50% line is yellow (not so good) and the 95 as red (bad) for goal purposes. These plots should be very useful for teaching purposes for professionals and patients, i.e. the steep slope and the plateau.  Their plots show the BP-ENa relationship extends down to maybe 20 mM Na per 24 hours. Is this an extension for drawing the curve or did they actually have 24 hr urines that were this low? 

Response: The extension of 24-hour sodium data points on both edges of the plots is actual. Nevertheless, the 5% and 95% vertical lines suggest very few data points outside of these vertical lines. The objective of using 5%ile, 50%ile, and 95%ile lines in the spline plots was to provide an impression of the distribution of data points in the spline plots. For example, the segment of spline plots after the 95%ile line is lengthy but contains only 5% of the data points, whereas the segment between the 5%ile and 95%ile lines includes 90% of the data points. 

We agree with the reviewer that providing information about the low or recommended intake, such as putting a verticle line showing the DASH recommended intake would be useful. However, we want to be careful in such an approach because we are not sure which method of completeness evaluation is accurate. There is certainly some bias associated with each of the complete urine sample assessment approaches since we did not compare these approaches with the gold-standard PABA excretion approach. Therefore, labeling may provide misinformation in the absence of comparison with the gold-standard PABA approach.  

  1. If “weather” plays a role in ENa I would expect that ENa might be related to average daily temperature which could be assessed by using weather data during the study.

Response: Thanks for the great suggestions. We agree with the reviewer that secondary meteorological data can be collected from the government bodies of Bangladesh. Such data are not publicly available, and therefore we are unable to use them for the current paper. Nevertheless, we will explore the possibilities to retrieve the temperature and humidity data to use in future.

  1. Their cubic spline plots also demonstrate a dose effect with what looks like a plateau. If one tests the effect only for those with an ENa above the 50th percentile is it linear or non-linear? It does appear that in some plots above the 50th they actually show a lower BP at very high ENa.  Perhaps these folks are ill (have a lower BP) due to an inability to retain Na? 

Responses: Thanks so much for your comments. We agree with the reviewer that the dose-response relationship for urine sodium excretion and systolic blood pressure relationship looks plateaued for both-sex combined plots above the 50th percentile. One of the possibilities includes the presence of blood pressure-lowering minerals (e.g., magnesium, potassium, and calcium) in saline water. In other analyses reported elsewhere, we identified drinking saline water is associated with not only a high intake of sodium but also higher intakes of magnesium, calcium, and potassium. We identified that urinary magnesium and calcium were associated with lower blood pressure, whereas urine sodium was associated with higher blood pressure. Secondly, there could be possible sex-specific salt and blood pressure interaction for different minerals, since the sex-specific spline plots suggest the plateau or decline of systolic blood pressure after the 50th percentile is only for men, not for women. We have included this discussion in the revised manuscript.

References:

  1. Wasserstein, R.L., A.L. Schirm, and N.A. Lazar, Moving to a world beyond “p< 0.05”. 2019, Taylor & Francis.

Reviewer 2 Report

Naser et al. conduct a comprehensive and well-conducted study that compares the sodium intake and systolic blood pressure (SBP) relationship from complete 24-hour urine samples determined by several methods. This study presents a correct design, a considerable sample size and a well-planned and performed analysis of the data. I consider that it can be very useful for the rest of researchers and clinicians working in this field. It is only necessary to make some minimal corrections:

  • Line 117: Do you use a commercial kit to perform the Jaffé method? otherwise, please explain the procedure in a simple way or include a reference in which it is explained.
  • Line 166: I think the ethical aspects should appear at the beginning of the methodology, not at the end.
  • Line 190: there is a mistake in the symbol ], it should be [

Author Response

### Reviewer 2

Naser et al. conduct a comprehensive and well-conducted study that compares the sodium intake and systolic blood pressure (SBP) relationship from complete 24-hour urine samples determined by several methods. This study presents a correct design, a considerable sample size and a well-planned and performed analysis of the data. I consider that it can be very useful for the rest of researchers and clinicians working in this field. It is only necessary to make some minimal corrections:

  • Line 117: Do you use a commercial kit to perform the Jaffé method? otherwise, please explain the procedure in a simple way or include a reference in which it is explained.

Response: We have included a reference in the method section.

  • Line 166: I think the ethical aspects should appear at the beginning of the methodology, not at the end.

Response: We have included the ethical sententence in the beginning of the method section.

  • Line 190: there is a mistake in the symbol ], it should be [
  • Response: We have corrected the symbol.

Reviewer 3 Report

In their study, Naser et al. investigate the relationship between 24 h urinary sodium excretion and systolic and diastolic blood pressure across a large sample from Bangladesh (n=10031 person-visits in n=1866 persons). The 24h urinary sodium excretion values were checked for completeness by several methods based on urinary creatinine excretion. The data were analyzed using multilevel linear models and restricted cubic spline plots to assess the difference in systolic and diastolic blood pressure for a 100 mmol increase in 24 h urinary sodium excretion.

The authors show a steep increase in systolic and diastolic BP with 24 h urinary sodium excretion until an inflection point from which on the relationship reaches a plateau (Fig. 2). The same data were also analyzed for sex-specific effects (Fig. 3). The authors also present linear models with values entered as tertiles and continuously (Table 3 /4). The relationship between 24 h urinary sodium excretion and BP were similar when only complete urinary samples were considered (around 1 mm Hg per 100 mmol Naper 24 h). As an exemption, inclusion of samples with urinary creatinine excretion between 15-25 mg/kg in males and 10-20 mg/kg in females lead to a higher relationship (roughly 2 mm Hg per 100 mmol Na per 24 h).

The presentation of the data is clear and the analyses are comprehensive. The discussion and conclusions are adequate.

I suggest the following changes:

  1. Abstract
    1. Please mention the number of all participants (line 27 “10031 BP and 24-hour urine sodium data from n=….persons”)
    2. Please add if the relationship between sys BP and 24h u-na is significantly higher in samples with adequate sex-specific mCER ranges
    3. The last sentence must be rephrased (line 39 two times “are”).
  2. Methods
    1. Drinking water can hardly be a source of dietary sodium since the sodium concentration is < 200 mg or <8 mmol per Liter. The source of dietary sodium is the food. If the authors still claim water to be the source of sodium, they should provide sodium concentration of the drinking water.
    2. In chapter 2.5., please indicate more clearly that the samples from incomplete collections were excluded.
  3. Results
    1. In Fig.1, please add he n number of each group
    2. Fig 3: are the sex-specific associations significantly different?
    3. In Table 3, please indicate the range of the tertiles
    4. In Table 4, please test the beta values for significance between each other. Please also indicate the n number of each row
    5. I suggest to repeat all analyses using mmol Na per 24 h per mg creatinine. This could be a superior approach with increased accuracy because it corrects for sampling errors.
  4. Discussion
    1. Please discuss why the criterion of sex-specific mCER yielded a higher beta value.

Author Response

### Reviewer 3

In their study, Naser et al. investigate the relationship between 24 h urinary sodium excretion and systolic and diastolic blood pressure across a large sample from Bangladesh (n=10031 person-visits in n=1866 persons). The 24h urinary sodium excretion values were checked for completeness by several methods based on urinary creatinine excretion. The data were analyzed using multilevel linear models and restricted cubic spline plots to assess the difference in systolic and diastolic blood pressure for a 100 mmol increase in 24 h urinary sodium excretion.

The authors show a steep increase in systolic and diastolic BP with 24 h urinary sodium excretion until an inflection point from which on the relationship reaches a plateau (Fig. 2). The same data were also analyzed for sex-specific effects (Fig. 3). The authors also present linear models with values entered as tertiles and continuously (Table 3 /4). The relationship between 24 h urinary sodium excretion and BP were similar when only complete urinary samples were considered (around 1 mm Hg per 100 mmol Naper 24 h). As an exemption, inclusion of samples with urinary creatinine excretion between 15-25 mg/kg in males and 10-20 mg/kg in females lead to a higher relationship (roughly 2 mm Hg per 100 mmol Na per 24 h).

The presentation of the data is clear and the analyses are comprehensive. The discussion and conclusions are adequate.

I suggest the following changes:

  1. Abstract
    1. Please mention the number of all participants (line 27 “10031 BP and 24-hour urine sodium data from n=….persons”)

Response: we have included the number of participants in the abstract.

  1. Please add if the relationship between sys BP and 24h u-na is significantly higher in samples with adequate sex-specific mCER ranges

Response: Thanks for your comments. We have provided the point estimates and confidence interval for urine sodium and blood pressure association for each method evaluated, which will help readers understand the statistical significance for each method. We refrained in explicitly mentioning which method was statistically significant as per the recent American Statistical Association recommendation[1].

  1. The last sentence must be rephrased (line 39 two times “are”).

Response: We have rephrased the last sentence of the abstract.

  1. Methods
    1. Drinking water can hardly be a source of dietary sodium since the sodium concentration is < 200 mg or <8 mmol per Liter. The source of dietary sodium is the food. If the authors still claim water to be the source of sodium, they should provide sodium concentration of the drinking water.

Response: Thanks for the comments. We agree with the reviewer that food is the main source of sodium intake. Nevertheless, several epidemiological studies in our study population highlighted that drinking saline water in the southwest coastal community is associated with sodium intake. We did not measure sodium concentration in water in our study; however, we have cited those references in the revised manuscript. We have also provided more context of water salinity and its seasonal variability in the revised manuscript.

  1. In chapter 2.5., please indicate more clearly that the samples from incomplete collections were excluded.

Response: We have clarified this in the revised manuscript.

  1. Results
    1. In Fig.1, please add he n number of each group

Response: We have included number of participants in each group in Figure 1.

  1. Fig 3: are the sex-specific associations significantly different?

Response: The restricted cubic spline plots illustrate the shape of associations and the linearity, but not the association's magnitude or significance. To overcome this limitation, we have also reported the tertile analyses, which provide the magnitude and significance of associations.

  1. In Table 3, please indicate the range of the tertiles

Response: We have included the ranges of tertiles in the revised manuscript in Table 3 footnote.

  1. In Table 4, please test the beta values for significance between each other. Please also indicate the n number of each row

Response: We are not sure what the reviewer reffered to by the test of beta values between each other. We have presented the point estimate and confidence interval for each beta value in Table 4. We have also presented the participants number each row in the revised manuscript.

  1. I suggest to repeat all analyses using mmol Na per 24 h per mg creatinine. This could be a superior approach with increased accuracy because it corrects for sampling errors.

Response: Thanks for your suggestion. We agree that in some scenarios, such as for spot urine metabolites, presenting urine metabolites per creatinine concentration can be a good approach to correct the sampling error. Nevertheless, we respectfully decline to repeat analyses considering the urine sodium-creatinine ratio for two reasons. First, all urine metabolites in this manuscript are from 24-hour urine excretions and were derived by multiplying the 24-hour urine volume. Therefore, we do not believe that such a sampling error is an issue for the current analyses. Secondly, but importantly, using urine metabolite-creatinine ratio in multivariate regression analyses can lead to bias. For instance, a urine metabolite itself may be unrelated to the outcome variable, but creatinine may be related to the outcome variable[2]. The urine metabolite could potentially achieve statistical significance only because of the use of urine metabolite-creatine ratio. Because age, sex, and body mass index all relate to urinary creatinine, including urine metabolite-creatinine ratio can also lead to a substantial multicollinearity problem in multivariate regression models[3].

  1. Discussion
    1. Please discuss why the criterion of sex-specific mCER yielded a higher beta value.

Response: Thanks for the comments. We have included the possible explanation of higher beta value for the sex-specific mCER method in the revised manuscript.